# OPTIMAL TRANSPORT, CYCLEGAN, AND PENALIZED LS FOR UNSUPERVISED LEARNING IN INVERSE PROBLEMS

## ABSTRACT

The penalized least squares (PLS) is a classic method to inverse problems, where a regularization term is added to stabilize the solution. Optimal transport (OT) is another mathematical framework for computer vision tasks that provides means to transport one measure to another at minimal cost. The cycle-consistent generative adversarial network (cycleGAN) is a recent extension of GAN to learn target distributions with less mode collapsing behavior. Although similar in that no supervised training is required, the algorithms look different, so the mathematical relationship between these approaches is not clear. In this article, we provide an important advance to unveil the missing link. Specifically, we reveal that a cycleGAN architecture can be derived as a dual formulation of the OT problem, if the consistency constraint of PLS is enforced as a regularization term of the OT problem. This suggests that cycleGAN can be considered stochastic generalization of classical PLS approaches. Our derivation is so general that various types of cycleGAN architectures can be easily derived by merely changing the transport cost. As proofs of concept, this paper provides a novel cycleGAN architecture for unsupervised learning in accelerated magnetic resonance imaging (MRI) and deconvolution microscopy problems, which confirm the efficacy and the flexibility of the theory.

## 1 INTRODUCTION

Inverse problems are ubiquitous in computer vision, biomedical imaging, scientific discovery, etc. In inverse problems, a noisy measurement $y \in \mathcal{Y}$ from an unobserved image $x \in \mathcal{X}$ is modeled by

$$y = \mathcal{H}x + w, \qquad (1)$$

where $w$ is the measurement noise, and $\mathcal{H} : \mathcal{X} \mapsto \mathcal{Y}$ is the measurement operator. In inverse problems originating from physics, the measurement operator is usually represented by an integral equation:

$$\mathcal{H}x(\boldsymbol{r}) \quad := \quad \int_{\mathbb{R}^d} h(\boldsymbol{r}, \boldsymbol{r}')x(\boldsymbol{r}')d\boldsymbol{r}', \quad \boldsymbol{r} \in \mathcal{D} \subset \mathbb{R}^d, \quad d = 2, 3, \qquad (2)$$

where $h(\boldsymbol{r}, \boldsymbol{r}'; x)$ is an integral kernel. Then, the inverse problem is formulated as an estimation problem of the unknown $x$ from the measurement $y$. It is well known that inverse problems are ill-posed. A classical strategy to mitigate the ill-posedness is the penalized least squares (PLS) approach:

$$\hat{x} = \arg \min_x c(x; y) := \|y - \mathcal{H}x\|^q + R(x) \qquad (3)$$

for $q \geq 1$, where $R(x)$ is a regularization (or penalty) function ($l_1$, total variation (TV), etc.) (Chaudhuri et al., 2014; Sarder & Nehorai, 2006; McNally et al., 1999). In some inverse problems, the measurement operator $\mathcal{H}$ is not well defined, so both the unknown operator $\mathcal{H}$ and the image $x$ should be estimated.

Recently, deep learning approaches with supervised training have become the mainstream approaches for inverse problems because of their excellent and ultra-fast reconstruction performance.

For example, in low-dose x-ray computed tomography (CT) denoising problems, a convolutional neural network (CNN) is trained to learn the relationship between the noisy image $y$ and the matched noiseless (or high-dose) label images $x$ (Kang et al., 2017). In the context of (3), the supervised neural network can be understood as directly learning the operation $\hat{x} = \arg\min_x c(x; y)$. Unfortunately, in many applications, matched label data are not available. Therefore, unsupervised training without matched reference data has become an important research topic.

Recently, the generative adversarial network (GAN) has attracted significant attention in the machine learning community by providing a way to generate target data distribution from random distribution (Goodfellow et al., 2014). In particular, Arjovsky et al. (2017) proposed the so-called Wasserstein GAN (W-GAN), which is closely related to the mathematical theory of optimal transport (OT) (Villani, 2008; Peyré et al., 2019). In OT, for two given probability measures supported on the $\mathcal{X}$ and $\mathcal{Y}$ spaces, one pays a cost for transporting one measure to another. Then, the minimization of the average transportation cost provides an unsupervised way of learning the transport map between the two measures. Unfortunately, these GAN approaches often generate artificial features due to mode collapsing, so cycle-consistent GAN (cycleGAN) (Zhu et al., 2017), which imposes one-to-one correspondence, has been extensively investigated (Kang et al., 2019; Lu et al., 2017). Recently, cycleGAN was generalized to bridge two different domains by generating a continuous sequence of intermediate domains flowing from one domain to the other (Gong et al., 2019).

Although classical PLS, OT, and cycleGAN share the commonality of unsupervised learning which does not require matched training data, there is no mathematical theory to systematically link these seemingly different approaches. Therefore, one of the main contributions of this paper is to unveil the missing link between these methods. In particular, we reveal that a cycleGAN architecture can be derived as a dual formulation of the OT problem when the consistency term in PLS is enforced as a regularization term for the OT problem. Moreover, this framework is so general that various cycle-GAN architectures can be easily obtained by changing the PLS cost. Accordingly, our framework provides a principled way of deriving unsupervised neural networks for various inverse problems.

As proofs of concept, we provide novel cycleGAN architectures for unsupervised learning in two physical inverse problems: accelerated MRI and deconvolution microscopy. In these applications, we show that only one CNN generator is required, which significantly reduces the training complexity. The experimental results confirm that the proposed unsupervised learning approaches can successfully provide accurate inversion results without any matched reference. All the proofs of the main theoretical results can be found in the Appendix.

## 2 RELATED WORKS

### 2.1 OPTIMAL TRANSPORT (OT)

OT compares two measures in a Lagrangian framework (Villani, 2008; Peyré et al., 2019). Formally, we say that $T : \mathcal{X} \mapsto \mathcal{Y}$ transports $\mu \in P(\mathcal{X})$ to $\nu \in P(\mathcal{Y})$, if

$$\nu(B) = \mu\left(T^{-1}(B)\right), \quad \text{for all } \nu\text{-measurable sets } B, \tag{4}$$

which is often simply represented by $\nu = T_{\#}\mu$, where $T_{\#}$ is often called the push-forward operator. Monge's original OT problem (Villani, 2008; Peyré et al., 2019) is then to find a transport map $T$ that transports $\mu$ to $\nu$ at the minimum total transportation cost:

$$\min_{T} \quad \mathbb{M}(T) := \int_{\mathcal{X}} c(x, T(x)) d\mu(x), \quad \text{subject to} \quad \nu = T_{\#}\mu$$

However, this is usually computationally expensive due to the nature of combinatorial assignment. Kantorovich relaxed the assumption to consider probabilistic transport that allows mass splitting from a source toward several targets (Villani, 2008; Peyré et al., 2019). Specifically, Kantorovich introduced a joint measure $\pi \in P(\mathcal{X} \times \mathcal{Y})$ and the associated cost $c(x, y), x \in \mathcal{X}, y \in \mathcal{Y}$ such that the original problem can be relaxed as

$$\min_{\pi} \quad \mathbb{K}(\pi) := \int_{\mathcal{X} \times \mathcal{Y}} c(x, y) d\pi(x, y) \tag{5}$$
$$\text{subject to} \quad \pi(A \times \mathcal{Y}) = \mu(A), \quad \pi(\mathcal{X} \times B) = \nu(B)$$

for all measurable sets $A \in \mathcal{X}$ and $B \in \mathcal{Y}$. Here, the last two constraints come from the observation that the total amount of mass removed from any measurable set has to equal the marginals (Villani,

2008; Peyré et al., 2019). Another important advantage of Kantorovich formulation is the dual formulation as stated in the following theorem:

**Theorem 1** (Kantorovich duality theorem). *(Villani, 2008, Theorem 5.10, p.57-p.59) Let $(\mathcal{X}, \mu)$ and $(\mathcal{Y}, \nu)$ be two Polish probability spaces (separable complete metric space) and let $c : \mathcal{X} \times \mathcal{Y} \to \mathbb{R}$ be a continuous cost function, such that $|c(x, y)| \leq c_{\mathcal{X}}(x) + c_{\mathcal{Y}}(y)$ for some $c_{\mathcal{X}} \in L^1(\mu)$ and $c_{\mathcal{Y}} \in L^1(\nu)$, where $L^1(\mu)$ denotes the set of 1-Lipschitz functions with the measure $\mu$. Then, there is duality:*

$$\min_{\pi \Pi(\mu, \nu)} \int_{\mathcal{X} \times \mathcal{Y}} c(x, y) d\pi(x, y) = \max_{\varphi \in L^1(\mu)} \left\{ \int_{\mathcal{X}} \varphi(x) d\mu(x) + \int_{\mathcal{Y}} \varphi^c(y) d\nu(y) \right\}$$

*and the above maximum is taken over the so-called* Kantorovich potential $\varphi$ *whose c-transform $\varphi^c(y) = \sup_x (c(x, y) - \varphi(x))$ is properly defined.*

## 2.2 PLS WITH DEEP LEARNING PRIOR

Recently, PLS frameworks using a deep learning prior have been extensively studied (Zhang et al., 2017; Aggarwal et al., 2018) thanks to their similarities to the classical regularization theory. The main idea of these approaches is to utilize a pre-trained neural network to stabilize the inverse solution. For example, in model based deep learning architecture (MoDL) (Aggarwal et al., 2018), the problem can be formulated as

$$\min_x c(x; y, \Theta, \mathcal{H}) = \|y - \mathcal{H}x\|^2 + \lambda \|x - Q_\Theta(x)\|^2 \tag{6}$$

for some regularization parameter $\lambda > 0$, where $Q_\Theta(x)$ is a pre-trained CNN with the network parameter $\Theta$ and the input $x$. This problem is usually solved in an alternating minimization framework:

$$x_{n+1} = \arg\min_x \|y - \mathcal{H}x\|^2 + \lambda \|x - z_n\|^2 \quad , \quad z_{n+1} = Q_\Theta(x_{n+1}) \tag{7}$$

Another type of inversion approach using a deep learning prior is the so-called deep image prior (DIP) (Ulyanov et al., 2018). Rather than using an explicit prior, the deep neural network architecture itself is used as a regularization by restricting the solution space:

$$\min_\Theta c(\Theta; y, \mathcal{H}) = \|y - \mathcal{H}Q_\Theta(z)\|^2 \tag{8}$$

where $z$ is a random vector. Then, the final solution becomes $x = Q_{\Theta^*}(z)$ with $\Theta^*$ being the estimated network parameters. Similarly, generative approaches (Van Veen et al., 2018; Bora et al., 2017; Wu et al., 2019) either estimate the random variable $z$ from (8) by fixing $\Theta$ or attempt to estimate both the random $z$ and the network weight $\Theta$.

## 3 MAIN CONTRIBUTIONS

One of the basic assumptions in the aforementioned PLS formulation with a deep learning prior is that the measurement $y$ is fixed and one is interested in finding the unknown $x$. In this paper, we relax the assumption to consider situations where both of them are random samples from joint probability measure $\pi(x, y)$. Specifically, we propose a new PLS cost function with a novel deep learning prior as follows:

$$c(x, y; \Theta, \mathcal{H}) = \|y - \mathcal{H}x\|^q + \lambda \|G_\Theta(y) - x\|^p \tag{9}$$

with $p, q \geq 1$, where the first two terms before the semi-colon are the random vectors. Here, the regularization parameter $\lambda$ takes care of the dimensional difference of $x$ and $y$, with $\lambda = 1$ for the same dimension, but for simplicity, in the rest of the paper, we assume $\lambda = 1$.

The new PLS cost in (9) has very unique properties. Suppose that the global minimum with $c(x, y; \Theta, \mathcal{H}) = 0$ exists. The corresponding global minimizer should then satisfy

$$y = \mathcal{H}x, \quad x = G_\Theta(y) \tag{10}$$

If this holds for all $y$, then $G_\Theta$ is an inverse operator of $\mathcal{H}$. In practice, due to the limited capacity of the neural network, the global minimum with $c(x, y; \Theta, \mathcal{H}) = 0$ may never be achieved. Even in this

case, thanks to the symmetric form of PLS, the cost in (9) has an important implication: the neural network $G_\Theta$ is now estimated by enforcing the consistency $y = \mathcal{H}x$ as a regularization term. This implies that the roles of the consistency and regularization terms in the PLS can be interchanged in our formulation. Another important advantage is that the unknown image $x$ can be obtained as an output of the feedforward neural network $G_\Theta(y)$ for the given measurement $y$. This makes the inversion procedure much simpler. One may say that the PLS cost function for the DIP in (8) can produce a different variation of unsupervised learning. However, the resulting formulation is not a feed-forward neural network. In fact, to obtain the unknown image $x$, an additional optimization step is required even after the neural network is trained. See Appendix C for more details.

Note that $\Theta, \mathcal{H}$ are the network and measurement system parameters that should be estimated. These parameters can be found by minimizing the average transport cost for all combinations of $x \in \mathcal{X}$ and $y \in \mathcal{Y}$ with respect to the joint measure $\pi(x, y)$:

$$\mathbb{T}(\Theta, \mathcal{H}) := \min_\pi \int_{\mathcal{X} \times \mathcal{Y}} c(x, y; \Theta, \mathcal{H}) d\pi(x, y) \tag{11}$$

where the minimum is taken over all joint distributions whose marginal distributions with respect to $X$ and $Y$ are $\mu$ and $\nu$, respectively. Below, we show that the average transportation cost in (11) has an interesting decomposition, which leads to the cycleGAN architecture.

**Lemma 1.** *If the mapping $G_\Theta : \mathcal{Y} \mapsto \mathcal{X}$ is single-valued, then the average transportation cost $\mathbb{T}(\Theta, \mathcal{H})$ in (11) can be decomposed as*

$$\mathbb{T}(\Theta, \mathcal{H}) = \ell_{cycle}(\Theta, \mathcal{H}) + \ell_{OT'}(\Theta, \mathcal{H}) \tag{12}$$

*where*

$$\ell_{cycle}(\Theta, \mathcal{H}) = \min_\pi \int_{\Gamma(G_\Theta) \cup \Gamma(\mathcal{H})} c(x, y; \Theta, \mathcal{H}) d\pi(x, y) \tag{13}$$

$$\ell_{OT'}(\Theta, \mathcal{H}) = \min_\pi \int_{\mathcal{X} \times \mathcal{Y} \setminus \Gamma(G_\Theta) \cup \Gamma(\mathcal{H})} c(x, y; \Theta, \mathcal{H}) d\pi(x, y) \tag{14}$$

*and $\Gamma(G_\Theta)$, $\Gamma(\mathcal{H})$ denote the graphs of $G_\Theta$, $\mathcal{H}$, respectively:*

$$\Gamma(G_\Theta) := \{(x, y) \in \mathcal{X} \times \mathcal{Y} \mid x = G_\Theta(y), \ y \in \mathcal{Y}\}, \quad \Gamma(\mathcal{H}) := \{(x, y) \in \mathcal{X} \times \mathcal{Y} \mid y = \mathcal{H}x, \ x \in \mathcal{X}\}$$

Note the nature of the sets $\Gamma(G_\Theta)$ and $\Gamma(\mathcal{H})$ and $\ell_{cycle}$. Since both $\Gamma(G_\Theta)$ and $\Gamma(\mathcal{H})$ are graphs of functions, they are low-dimensional manifolds embedded in the ambient space $\mathcal{X} \times \mathcal{Y}$. In this sense, their relative measures become zero and $\ell_{cycle}$ seems to be a negligible part of the whole $\mathbb{T}$. This term, however, can contribute a significant portion of the total loss $\mathbb{T}$ depending on the existence of the singularity in the distribution. We defer the discussion to after the following proposition, which explains why we name the first term $\ell_{cycle}$.

**Proposition 1.**

$$\ell_{cycle}(\Theta, \mathcal{H}) := \min_\pi \int_{\Gamma(G_\Theta) \cup \Gamma(\mathcal{H})} c(x, y; \Theta, \mathcal{H}) d\pi(x, y)$$

$$= \int_{\Gamma(G_\Theta) \cup \Gamma(\mathcal{H})} c(x, y; \Theta, \mathcal{H}) \pi^*(dx, dy)$$

$$= \int_{\mathcal{X}} \|x - G_\Theta(\mathcal{H}x)\|^p \rho(x) \mu(dx) + \int_{\mathcal{Y}} \|y - \mathcal{H}G_\Theta(y)\|^q \sigma(y) \nu(dy) \tag{15}$$

*where $\pi^*$ denotes the optimum measure and $\rho$ and $\sigma$ are measurable functions such that $0 \leq \rho(x) \leq 1$ $\mu$-a.e., and $0 \leq \sigma(y) \leq 1$ $\nu$-a.e.*

The remaining term in (12), which corresponds to the GAN term, can be obtained as follows:

**Proposition 2.** *For $p = q = 1$, the cost $\ell_{OT'}(\Theta, \mathcal{H})$ in (14) can be computed as*

$$\ell_{OT'}(\Theta, \mathcal{H}) = \max_\varphi \int_{\mathcal{X}} \varphi(x)[1 - \rho(x)] d\mu(x) - \int_{\mathcal{Y}} \varphi(G_\Theta(y))[1 - \sigma(y)] d\nu(y) \tag{16}$$

$$+ \max_\psi \int_{\mathcal{Y}} \psi(y)[1 - \sigma(y)] d\nu(y) - \int_{\mathcal{X}} \psi(\mathcal{H}x)[1 - \rho(x)] d\mu(x) \tag{17}$$

*where $\varphi$ and $\psi$ are 1-Lipschitz Kantorovich potential functions.*

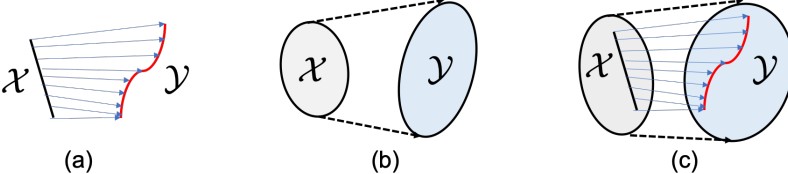

Figure 1: (a) Supervised learning with matched references, (b) OT with absolute continuous measures, and (c) OT having measures with singularities. CycleGAN corresponds to scenario (c).

The proof of Propositions 1 and 2 can be found in Appendix B. The proof idea comes from the computation of $c(x, y; \Theta, \mathcal{H})$ on subsets $\Gamma(G_\Theta), \Gamma(\mathcal{H})$, and $\Gamma(G_\Theta) \cap \Gamma(\mathcal{H})$ by considering the singularity of the optimum measure $\pi^*$ on these sets. In this regard, it is important to know how singularity of $\pi^*$ is distributed on these manifolds. If the targets are perfectly matched as shown in Fig. 1(a), $\sigma(y) := \pi^*(\{x = G_\Theta(y)\}|y) = 1; \nu$-a.e so that $\pi^*(\cdot|y)$ is distributed like a point mass of magnitude $\sigma(y)$ on $\{x = G_\Theta(y)\}$. This also leads to $\rho(x) := \pi^*(\{y = \mathcal{H}x\}|x) = 1; \mu$-a.e. In this case, $\ell_{OT'} = 0$ and $G_\Theta(y)$ is a perfect inverse to compute $x$. On the other hand, if the joint distribution $\pi(x, y)$ is absolutely continuous as shown in Fig. 1(b), then $\rho(x) = 0; \mu$-a.e. and $\sigma(y) = 0; \nu$-a.e. In this case, $\ell_{cycle} = 0$ and the contribution of the cycle consistency term can be neglected. The most interesting case arises when the joint measure $\pi(x, y)$ has significant singularities on the manifold $\Gamma(G_\Theta) \cup \Gamma(\mathcal{H})$ so that $0 < \sigma(y) = \pi^*(\{x = G_\Theta(y)\}|y) < 1$ (see Fig. 1(c)). In fact, unsupervised learning applies to such situations where once the mappings are found, a significant portion of the data in $\mathcal{X}$ and $\mathcal{Y}$ can be paired with the mappings, even though unpaired data remain. This is when the cycle consistency terms plays an important role in unsupervised learning. Then, $G_\Theta(y)$ is a desirable approximation of $x$ with high probability.

Now, the final step is implementing the Kantorovich potential using CNNs with parameters $\Phi$ and $\Xi$, i.e. $\varphi := \varphi_\Phi$ and $\psi := \psi_\Xi$. By collecting all terms in the transportation cost $\mathbb{T}(\Theta, \mathcal{H})$, the dual formulation results in a cycleGAN cost function:

$$\min_{\Theta, \mathcal{H}} \mathbb{T}(\Theta, \mathcal{H}) = \min_{\Theta, h} \max_{\Phi, \Xi} \ell(\Theta, \mathcal{H}; \Phi, \Xi) \tag{18}$$

where

$$\ell(\Theta, \mathcal{H}; \Phi, \Xi) = \ell_{cycle}(\Theta, \mathcal{H}) + \ell_{GAN}(\Theta, \mathcal{H}; \Phi, \Xi) \tag{19}$$

where $\ell_{cycle}(\Theta, \mathcal{H})$ denotes the cycle-consistency loss in (15) and $\ell_{GAN}(\Theta, h; \Phi, \Xi)$ is the GAN loss given by:

$$\ell_{GAN}(\Theta, \mathcal{H}; \Phi, \Xi) = \int_{\mathcal{X}} \varphi_\Phi(x)[1 - \rho(x)]d\mu(x) - \int_{\mathcal{Y}} \varphi_\Phi(G_\Theta(y))[1 - \sigma(y)]d\nu(y) \tag{20}$$

$$+ \int_{\mathcal{Y}} \psi_\Xi(y)[1 - \sigma(y)]d\nu(y) - \int_{\mathcal{X}} \psi_\Xi(\mathcal{H}x)[1 - \rho(x)]d\mu(x)$$

Here, the Kantorovich 1-Lipschitz potentials $\varphi := \varphi_\Phi$ and $\psi := \psi_\Xi$ correspond to the W-GAN discriminators. Specifically, $\varphi_\Phi$ tries to find the difference between the true image $x$ and the generated image $G_\Theta(y)$, whereas $\psi := \psi_\Xi$ attempts to find the fake measurement data that are generated by the synthetic measurement procedure $\mathcal{H}x$. Furthermore, if $\rho$ and $\sigma$ are constant almost everywhere, they become simple scaling factors for the cycle-consistency terms:

$$\ell_{cycle}(\Theta, \mathcal{H}) = \rho \int_{\mathcal{X}} \|x - G_\Theta(\mathcal{H}x)\|^p d\mu(x) + \sigma \int_{\mathcal{Y}} \|y - \mathcal{H}G_\Theta(y)\|^q d\nu(y) \tag{21}$$

Note that our formulation using (19) with (20) and (21) has many unique features compared to the standard cycleGAN (Zhu et al., 2017).

- The standard cycleGAN (Zhu et al., 2017) usually requires two generators in the form of deep neural networks. On the other hand, our formulation often requires a single generator using a deep neural network, if the other generator is given by the forward model $\mathcal{H}x$. This significantly reduces the number of weights, which makes the training much more stable. This can be viewed as the consistency term from the forward model working as a regularization term for OT.

- When the solution of the dual problem achieves the global minimum with diminishing cost, then the dual solution $x = G_\Theta(y)$ is equivalent to the primal solution (10), validating the dual approach. This is not the case in the conventional cycleGAN with two deep generators.

- While we only consider $p = q = 1$ due to the simple $c$-transform $\varphi^c(x) = -\varphi(x)$ for 1-Lipschitz $\varphi$, the use of the general PLS cost would be interesting, and it may lead to an interesting variation of the cycleGAN architecture. This could be done using a regularized version of OT (Peyré et al., 2019).

- The parameters $\rho(x)$ and $\sigma(y)$ in (1) and (20) originate from the singularity of the optimal joint measure. However, the estimation of $\rho(x)$ and $\sigma(y)$ is not feasible in practice, so the constant hyperparameters in (21) are usually used, and the hyper parameters should be selected by trial and error, which is the same as in the standard cycleGAN. Nevertheless, the probability interpretation may lead to a more simplified search for the hyper-parameters.

## 4 APPLICATIONS TO INVERSE PROBLEMS

### 4.1 ACCELERATED MRI

In accelerated magnetic resonance imaging (MRI), the goal is to recover high-quality MR images from sparsely sampled $k$-space data to reduce the acquisition time. This problem has been extensively studied using compressed sensing (Lustig et al., 2007), but recently, deep learning approaches have been the main research interest due to their excellent performance and significantly reduced run-time complexity (Hammernik et al., 2018; Han & Ye, 2019). A standard method for neural network training for accelerated MRI is based on supervised learning, where the MR images from fully sampled $k$-space data are used as references and subsampled $k$-space data are used as the input for the neural network. Unfortunately, in accelerated MRI, high-resolution fully sampled $k$-space data are very difficult to acquire due to the long scan time. Therefore, the need for unsupervised learning without matched reference data is increasing.

In accelerated MRI, the forward measurement model can be described as

$$\hat{x} = \mathcal{P}_\Omega \mathcal{F} x + w \tag{22}$$

where $\mathcal{F}$ is the 2-D Fourier transform and $\mathcal{P}_\Omega$ is the projection to $\Omega$ that denotes $k$-space sampling indices. To implement every step of the algorithm as image domain processing, (22) can be converted to the image domain forward model by applying the inverse Fourier transform:

$$y = \mathcal{F}^{-1} \mathcal{P}_\Omega \mathcal{F} x + \mathcal{F}^{-1} w \tag{23}$$

This results in the following cost function for the PLS formulation:

$$c(x, y; \Theta) = \|y - \mathcal{F}^{-1} \mathcal{P}_\Omega \mathcal{F} x\| + \|G_\Theta(y) - x\| \tag{24}$$

where we set $\lambda = 1$ in (9) since the dimensions of $x$ and $y$ are the same. Usually, the sampling mask $\Omega$ is known so that the forward mapping for the inverse problem is deterministic.

The schematic diagram of the associated cycleGAN architecture is illustrated in Fig. 2, whose generator and discriminator architectures are shown in Fig. 6 in Appendix D. Note that we just need a

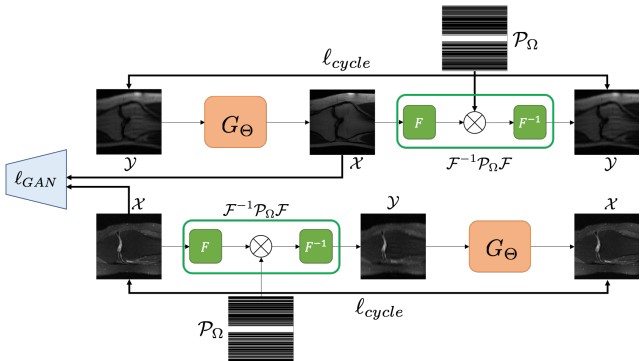

Figure 2: Proposed cycleGAN architecture for accelerated MRI with 1-D downsampling patterns.

Table 1: Quantitative comparison for various algorithms on 500 test sets of fastMRI data.

| Metric | Input ($\times 4$) | Supervised Learning | Conventional CycleGAN | Proposed Method |
|---|---|---|---|---|
| PSNR (dB) | 26.8056 | 27.9609 | 25.0092 | **28.1704** |
| SSIM | 0.6419 | 0.6419 | 0.4689 | **0.6550** |

single generator and discriminator, as the mapping from the clean to aliased images is deterministic for a given sampling pattern. As for the loss function, we use (19) with the $\sigma = \rho = 0.5$ for the cycle consistency term in (21). For GAN implementation, we use the W-GAN in (20) with the Lipschitz penalty loss (Gulrajani et al., 2017) to ensure that the Kantorovich potential becomes 1-Lipschitz. The detailed description of the training procedure is provided in Appendix D. The reconstruction results in Fig. 3 and quantitative comparison results for all test sets in Table 1 clearly show that the proposed unsupervised learning method using dedicated cycleGAN successfully recovered fine details without matched references. Moreover, the performance is even better than that of the standard cycleGAN and is comparable with that of supervised learning.

## 4.2 Deconvolution Microscopy

Deconvolution microscopy is extensively used to improve the resolution of widefield fluorescent microscopy. However, classical deconvolution approaches require the measurement or estimation of the point spread function (PSF) and are usually computationally expensive. Recently, CNN approaches have been extensively studied as fast and high-performance alternatives. Unfortunately, the CNN approaches usually require matched high-resolution images for supervised training. Therefore, we are interested in developing unsupervised deep neural networks for deconvolution microscopy. Mathematically, a blurred measurement can be described as

$$y \quad = \quad h * x + w \,, \tag{25}$$

where $h$ is the PSF. Here, we consider the blind deconvolution problem where both the unknown PSF $h$ and the image $x$ should be estimated. This results in the following cost function for the PLS

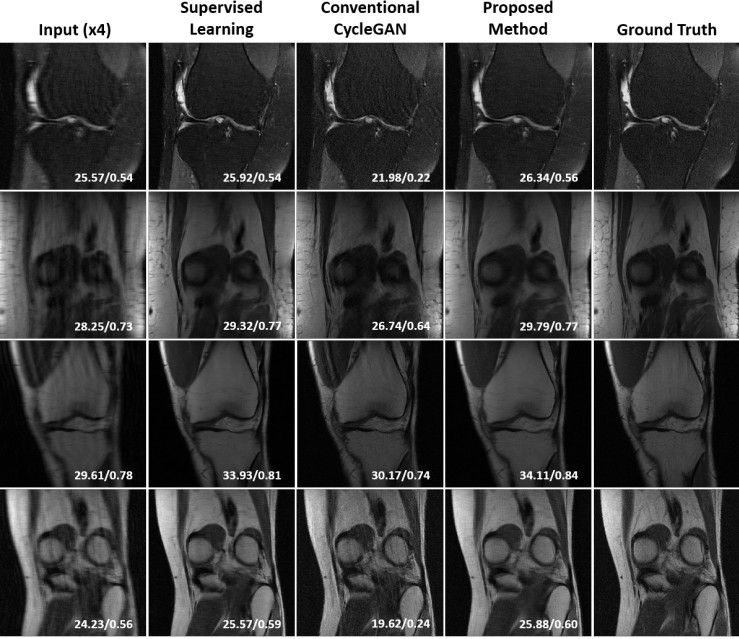

Figure 3: Unsupervised learning results for accelerated MRI using proposed cycleGAN. The values in the corners are PSNR/SSIM values for each image. Here, PSNR $= 20 \log_{10} (n \|x^*\|_\infty / \|x - x^*\|_2)$ and SSIM $= (2\mu_x \mu_{x^*} + c_1)(2\sigma_{xx^*} + c_2)/(\mu_x^2 + \mu_{x^*}^2 + c_1)(\sigma_x^2 + \sigma_{x^*}^2 + c_2)$, where $x$ and $x^*$ denote the reconstructed images and ground truth, respectively, $n$ is the number of pixels, $\mu_x$ is an average of $x$, $\sigma_x^2$ is a variance of $x$ and $\sigma_{xx^*}$ is a covariance of $x$ and $x^*$, and $c_1, c_2$ are two variables to stabilize the division.

formulation:

$$c(x, y; \Theta, h) = \|y - h * x\| + \|G_\Theta(y) - x\|. \tag{26}$$

The corresponding cycleGAN architecture is illustrated in Fig. 4. In contrast to the conventional cycleGAN approaches that require two generators, the proposed cycleGAN approach needs only a single generator, and the blur image can be generated using a linear convolution layer corresponding to the PSF, which significantly improves the robustness of network training. However, we still need two discriminators, since both linear and deep generators should be estimated. The detailed descriptions of the network architecture and training are given in Appendix D.

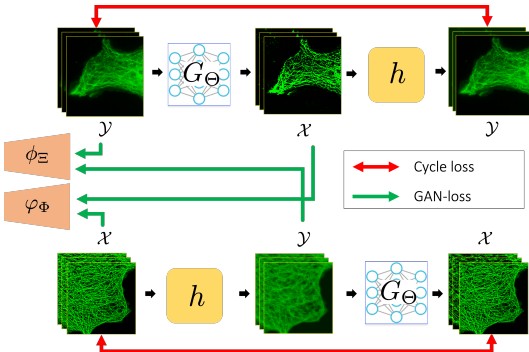

Figure 4: Proposed cycleGAN architecture with a blur kernel for deconvolution microscopy. Here, $G_\Theta$ denotes the CNN-based low-resolution to high-resolution generator. The blur generator is composed of a linear blur kernel $h$.

Fig. 5 shows lateral views of the deconvolution results of microtubule samples by various methods. Here, input images are degraded by blur and noise. The supervised learning and the standard cycleGAN with two generators showed better contrast and removed blur; however, the structural continuity was not preserved. On the other hand, in our cycleGAN approach, blur and noise were successfully removed, and the continuity of the structure was preserved. The quantitative comparison with known ground-truth data is provided in Appendix D, which clearly confirms the superiority of the proposed method,

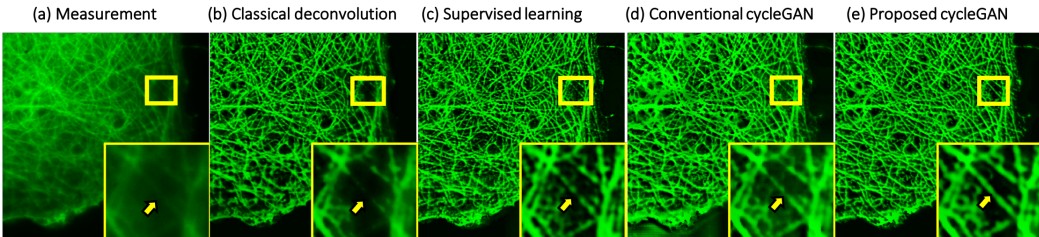

Figure 5: Comparison of reconstruction results by various methods: (a) blurred image measurements, (b) classical deconvolution method, (c) supervised learning, (d) conventional cycleGAN, and (e) proposed cycleGAN. The regions of interest (marked yellow) show enlargements of the areas marked in yellow.

## 5 CONCLUSIONS

In this paper, we presented a general design principle for a cycleGAN architecture for various inverse problems. Specifically, the proposed architecture was derived as a dual formulation of an OT problem that enforces the data consistency of PLS as a regularization term for the OT problem. As proofs of concept, we designed novel cycleGAN architectures for accelerated MRI and deconvolution microscopy examples, providing accurate reconstruction results without any matched reference data. Given the generality of our design principle, we believe that our method can be an important platform for unsupervised learning for inverse problems.

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

## A  ADDITIONAL THEOREMS AND LEMMA

**Theorem 2** (Optimality is inherited by restriction). *(Villani, 2008, Theorem 4.6, p.46) Under the same assumption of Kantorovich duality theorem, let $\pi^*$ be an optimal transport plan, and $\tilde{\pi}$ be a nonnegative measure on $\mathcal{X} \times \mathcal{Y}$, such that $\tilde{\pi} \leq \pi^*$ and $\tilde{\pi}[\mathcal{X} \times \mathcal{Y}] > 0$. Let $\pi' := \frac{\tilde{\pi}}{\tilde{\pi}[\mathcal{X} \times \mathcal{Y}]}$, $\mu', \nu'$ be the maginals of $\pi'$. Then optimality is inherited to its restriction, i.e., $\pi'$ is an optimal transference plan between $\mu'$ and $\nu'$.*

**Theorem 3** (Restriction for the Kantorovich duality). *(Villani, 2008, Theorem 5.19, p.75-p.76) Under the same assumption of Kantorovich duality theorem, let $\pi^*$ be an optimal transport plan and $\varphi^*$ be an optimal potential that attain the maximum, and $\tilde{\pi}$ be a measure on $\mathcal{X} \times \mathcal{Y}$, such that $0 \leq \tilde{\pi} \leq \pi^*$ and $Z := \tilde{\pi}[\mathcal{X} \times \mathcal{Y}] > 0$. Let $\pi' = \tilde{\pi}/Z$, and $\mu', \nu'$ be the marginals of $\pi'$. Then there exists $\varphi'$ such that $\varphi' = \varphi^*$ on $\text{proj}_{\mathcal{X}}(\text{supp}(\pi^*))$, and $\varphi'$ solves the dual Kantorovich problem between $(\mathcal{X}, \mu')$ and $(\mathcal{Y}, \nu')$.*

We are now ready to provide our duality result for the restricted measure.

**Lemma 2.** *Let*

$$
\begin{aligned}
\ell_{OT'} &:= \min_{\pi} \int_{\mathcal{X} \times \mathcal{Y} \setminus (A \cup B)} c(x, y) d\pi(x, y) \\
\ell_{OT} &:= \min_{\pi} \int_{\mathcal{X} \times \mathcal{Y}} c(x, y) d\pi(x, y)
\end{aligned}
$$

*Then, we can replace $\ell_{OT'}$ with $\ell_{OT}$ in the sense that both of them share the same optimal Kantorovich potential. More specifically, if $\varphi^*$ is the optimal Kantorovich potential, which satisfies*

$$\ell_{OT} = \int_{\mathcal{X}} \varphi^*(x)d\mu(x) + \int_{\mathcal{Y}} (\varphi^*)^c(y)d\nu(y),$$

*then*

$$\ell_{OT'} = \int_{\mathcal{X}} \varphi^*(x)[1 - \rho(x)]d\mu(x) + \int_{\mathcal{Y}} (\varphi^*)^c(y)[1 - \sigma(y)]d\nu(y).$$

*Proof.* This is a corollary of Theorem 2 and 3 by taking $\tilde{\pi} = \pi^*|_{\mathcal{X} \times \mathcal{Y} \setminus (\Gamma(G_\Theta) \cup \Gamma(\mathcal{H}))}$. Then, $\tilde{\pi}$ is nonnegative measure on $\mathcal{X} \times \mathcal{Y}$. Use the same notation for $Z, \mu', \nu'$ in Theorem 2 and 3. When $Z = 0$, $\ell_{OT'} = 0$, and $\mu' = 0, \nu' = 0$, so the statement trivially holds. Suppose $Z > 0$. Then by Theorem 3, there exists $\varphi'$ such that $\varphi' = \varphi^*$ on $\mathrm{proj}_{\mathcal{X}}(\mathrm{supp}(\pi^*))$, and $\varphi'$ solves the dual Kantorovich problem between $(\mathcal{X}, \mu')$ and $(\mathcal{Y}, \nu')$.

$$\ell_{OT'}/Z = \int_{\mathcal{X} \times \mathcal{Y}} c(x, y; \Theta, \mathcal{H}) \frac{d\pi(x, y)}{Z}$$
$$= \int_{\mathcal{X}} \varphi^*(x)d\mu'(x) + \int_{\mathcal{Y}} (\varphi^*)^c(y)d\nu'(y)$$

The remaining part of the proof is to show the relationship that $\mu' = \frac{1-\rho}{Z}\mu$ and $\nu' = \frac{1-\sigma}{Z}\nu$. From the proof of Proposition 1, the marginals of $\pi^*|_{\Gamma(G_\Theta) \cup \Gamma(\mathcal{H})}$ are $\rho\mu$ and $\sigma\nu$. Therefore, the marginals of $\tilde{\pi} = \pi^* - \pi^*|_{\Gamma(G_\Theta) \cup \Gamma(\mathcal{H})}$ are $\mu - \rho\mu$ and $\nu - \sigma\nu$ so that $Z\mu' = (1 - \rho)\mu$ and $Z\nu' = (1 - \sigma)\nu$.

$$\therefore \ell_{OT'} = \int_{\mathcal{X}} \varphi^*(x)[1 - \rho(x)]d\mu(x) + \int_{\mathcal{Y}} (\varphi^*)^c(y)[1 - \sigma(y)]d\nu(y)$$

$\square$

# B    PROOF OF MAIN RESULTS

## B.1    PROOF OF LEMMA 1

*Proof.* From Theorem 2, we extract information about two separated integrals. For some optimal transportation plan $\pi^*$, the restrictions $\pi^*|_{\Gamma(G_\Theta) \cup \Gamma(\mathcal{H})}$ and $\pi^*|_{\mathcal{X} \times \mathcal{Y} \setminus (\Gamma(G_\Theta) \cup \Gamma(\mathcal{H}))}$ acquire optimality. Therefore,

$$\mathbb{T}(\Theta, \mathcal{H}) = \int_{\mathcal{X} \times \mathcal{Y}} c(x, y; \Theta, \mathcal{H})d\pi^*(x, y)$$
$$= \int_{A \cup B} c(x, y; \Theta, \mathcal{H})d\pi^*(x, y) + \int_{\mathcal{X} \times \mathcal{Y} \setminus (A \cup B)} c(x, y; \Theta, \mathcal{H})d\pi^*(x, y)$$
$$= \ell_{cycle}(\Theta, \mathcal{H}) + \ell_{OT'}(\Theta, \mathcal{H})$$

This concludes the proof. $\square$

## B.2    PROOF OF PROPOSITION 1

*Proof.* By virtue of the definitions of the cost function and the sets $\Gamma(G_\Theta)$ and $\Gamma(\mathcal{H})$, we have

$$c(x, y; \Theta, \mathcal{H}) = \|y - \mathcal{H}G_\Theta(y)\|^q \text{ on } \Gamma(G_\Theta),$$
$$c(x, y; \Theta, \mathcal{H}) = \|x - G_\Theta(\mathcal{H}x)\|^p \text{ on } \Gamma(\mathcal{H}),$$
$$c(x, y; \Theta, \mathcal{H}) = 0 \quad \text{ on } \Gamma(G_\Theta) \cap \Gamma(\mathcal{H}).$$

Hence,

$$\int_{\Gamma(G_\Theta) \cup \Gamma(\mathcal{H})} c(x, y; \Theta, \mathcal{H})d\pi^*(x, y) = \int_{\Gamma(\mathcal{H})} \|x - G_\Theta(\mathcal{H}x)\|^p d\pi^*(x, y) + \int_{\Gamma(G_\Theta)} \|y - \mathcal{H}G_\Theta(y)\|^q d\pi^*(x, y)$$

Furthermore, we use disintegration theorem (Simmons, 2012) to split the joint measure $\pi^*$ as follows:

$$d\pi^*(x, y) = \pi^*(dy|x)\mu(dx) = \pi^*(dx|y)\nu(dy),$$

where $\pi^*(\mathcal{Y}|x) = 1$ $\mu$-a.e. and $\pi^*(\mathcal{X}|y) = 1$ $\nu$-a.e. Then, we have

$$\int_{\Gamma(\mathcal{H})} \|x - G_\Theta(\mathcal{H}x)\|^p d\pi^*(x, y) = \int_{\Gamma(\mathcal{H})} \|x - G_\Theta(\mathcal{H}x)\|^p \pi^*(dy|x)\mu(dx)$$

$$= \int_\mathcal{X} \int_{y=\mathcal{H}x} \|x - G_\Theta(\mathcal{H}x)\|^p \pi^*(dy|x)\mu(dx)$$

$$= \int_\mathcal{X} \|x - G_\Theta(\mathcal{H}x)\|^p \int \mathbf{1}_{y=\mathcal{H}x} \pi^*(dy|x)\mu(dx)$$

$$= \int_\mathcal{X} \|x - G_\Theta(\mathcal{H}x)\|^p \rho(x)\mu(dx)$$

where $\mathbf{1}_S$ denotes the indicator function for the set $S$ and $\rho(x) = \int \mathbf{1}_{y=\mathcal{H}x} \pi^*(dy|x)$. In a similar fashion, with $\sigma(y) = \int \mathbf{1}_{x=G_\Theta(y)} \pi^*(dx|y)$, we have

$$\int_{\Gamma(G_\Theta)} c(x, y; \Theta, \mathcal{H})d\pi^*(x, y) = \int_\mathcal{Y} \|y - \mathcal{H}G_\Theta(y)\|^q \sigma(y)\nu(dy) .$$

This concludes the proof. □

### B.3 PROOF OF PROPOSITION 2

*Proof.* This is a just simple corollary of the original Kantorovich's duality formulation proof and the classical results of optimal transport on the restricted measure. We start with the first term of $l_{OT}(\Theta, \mathcal{H})$ in (14). Specifically, we have

$$\int_{\mathcal{X}\times\mathcal{Y}\backslash(\Gamma(G_\Theta)\cup\Gamma(\mathcal{H}))} \|G_\Theta(y) - x\|^p \pi^*(dx, dy) = \max_\varphi \int_\mathcal{X} \varphi(x)\mu'(dx) + \int_\mathcal{Y} \varphi^c(G_\Theta(y))\nu'(dy)$$

where $\mu'$ and $\nu'$ are marginals of the restriction of optimal transportation plan on the restricted set $\mathcal{X} \times \mathcal{Y} \setminus (\Gamma(G_\Theta) \cup \Gamma(\mathcal{H}))$. Now, using Lemma 2 in Appendix, we have

$$\int_{\mathcal{X}\times\mathcal{Y}\backslash(\Gamma(G_\Theta)\cup\Gamma(\mathcal{H}))} \|G_\Theta(y) - x\|\pi^*(dx, dy)$$

$$= \max_\varphi \int_\mathcal{X} \varphi(x)[1 - \rho(x)]\mu(dx) + \int_\mathcal{Y} \varphi^c(G_\Theta(y))[1 - \sigma(y)]\nu(dy)$$

$$= \max_{\varphi\in L^1(\mu)} \int_\mathcal{X} \varphi(x)[1 - \rho(x)]\mu(dx) - \int_\mathcal{Y} \varphi(G_\Theta(y))[1 - \sigma(y)]\nu(dy)$$

for the non-restricted marginals $\mu$ and $\nu$, where for the last equality we use $\varphi^c = -\varphi$ for $p = 1$ when $\varphi$ is 1-Lipschitz (Villani, 2008). Using the same technique, the second term in (14) can be computed as

$$\int_{\mathcal{X}\times\mathcal{Y}\backslash(\Gamma(G_\Theta)\cup\Gamma(\mathcal{H}))} \|y - \mathcal{H}x\|d\pi(x, y)$$

$$= \max_{\psi\in L^1(\nu)} \int_\mathcal{Y} \psi(y)[1 - \sigma(y)]d\nu(y) - \int_\mathcal{X} \psi(\mathcal{H}x)[1 - \rho(x)]d\mu(x).$$

By collecting terms, we conclude the proof. □

## C OTHER VARIANTS OF UNSUPERVISED LEARNING

Here, we derive the dual optimal transport formulation when the PLS cost with deep learning prior as in (8) is used as the transportation cost for the OT problem. Specifically, the transportation cost of the OT problem with respect to the DIP cost in (8) becomes

$$\min_\Theta c(y, z; \Theta, \mathcal{H}) = \|y - \mathcal{H}Q_\Theta(z)\| \tag{27}$$

where we consider $y$ and $z$ as random variable and we use non-squared norm for simplicity. Then, the corresponding OT problem can be dualized as follows:

$$\min_{\Theta} \min_{\pi} \int_{\mathcal{X} \times \mathcal{Z}} \|y - \mathcal{H}G_{\Theta}(z)\| d\pi(x, z) = \min_{\Theta} \max_{\psi \in L^1(\nu)} \int_{\mathcal{Y}} \psi(y) d\nu(y) - \int_{\mathcal{X}} \psi(\mathcal{H}G_{\Theta}(z)) d\eta(z).$$

Although this is a nice way of pretraining deep image prior model using unmatched training data set, the final image estimate still comes from the following optimization problem:

$$x = G_{\Theta}(z^*) \quad \text{where} \quad z^* = \arg\min_{z} \|y - \mathcal{H}G_{\Theta^*}(z)\|$$

where $\Theta^*$ is the estimated network parameters from previous training step. This is equivalent to deep generator model (Bora et al., 2017). Therefore, this is not useful in designing a feed-forward neural network in an unsupervised manner.

## D  EXPERIMENTAL DETAILS

### D.1  ACCELERATED MRI

We use dataset for fastMRI challenge (Zbontar et al., 2018) for our experiments. This dataset is composed of MR images of knees. We extracted 3500 MR images from fastMRI single coil validation set. Then, 3000 slices are used for training/evaluation, and 500 slices are used for test. These MR images are fully sampled images, so we make undersampled images by a randomly subsampling k-space lines. The acceleration factor is four, and autocalibration signal (ACS) region contains 4% of k-space lines. Each slice is normalized by standard deviation of the magnitude of each slice. To handle complex values of data, we concatenate real and imaginary values along the channel dimension. Each slice has different size, so we use only single batch. The images are center-cropped to the size of $320 \times 320$, and then the peak signal-to-noise ratio (PSNR) and structural similarity index (SSIM) values are calculated.

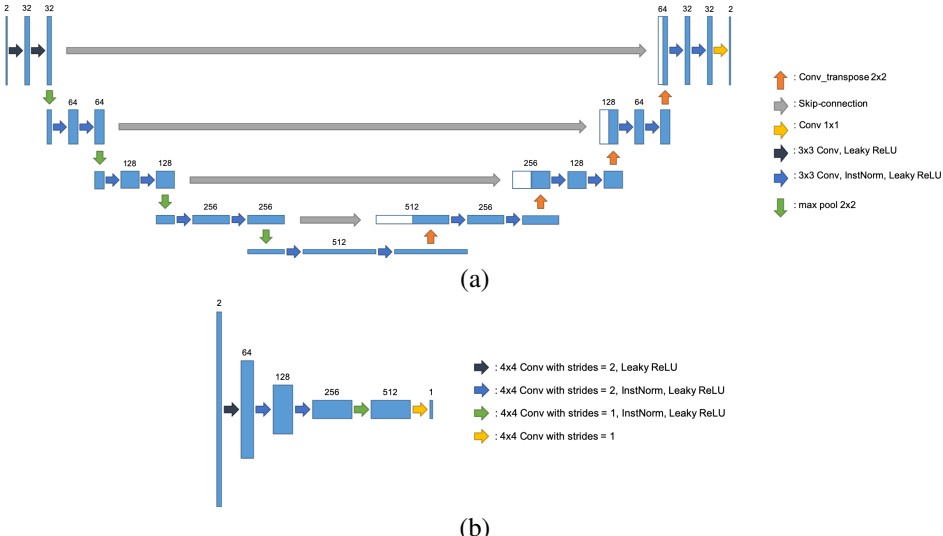

Figure 6: Proposed network architecture for (a) generator and (b) discriminator in accelerated MRI.

We use U-Net generator to reconstruct fully sampled MR images from undersampled MR images as shown in Fig. 6(a). Our generator consists of $3 \times 3$ convolution, Instance normalization, and leaky ReLU operation. Also, there are skip-connection and pooling layers. At the last convolution layer, we do not use any operation. Our discriminator is same as the discriminator of original CycleGAN. We use PatchGAN (Isola et al., 2017), so the discriminator classifies inputs at patch scales. The discriminator also consists of convolution layer, instance normalization, and leaky ReLU operation as shown in Fig. 6(b). We use Adam optimizer to train our network, with momentum parameters $\beta_1 = 0.5, \beta_2 = 0.9$, and learning rate of 0.0001. The discriminator is updated 5 times for every generator updates. We use batch size of 1, and trained our network during 100 epochs. Our code was implemented by TensorFlow.

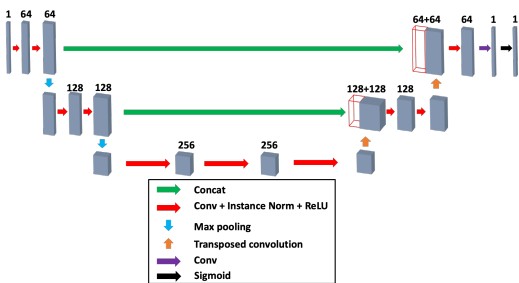

Figure 7: A modified 3D U-net architecture for our high-resolution image generator in the deconvolution microscopy problem.

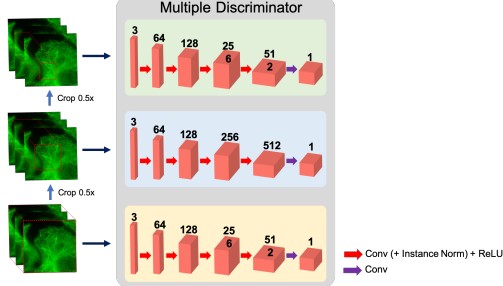

Figure 8: Multi-PatchGANs discriminator architecture in the deconvolution microscopy problem.

## D.2 DECONVOLUTION MICROSCOPY

The network architecture of the high resolution image generator $G_\Theta$ from the low-resolution image is a modified 3D-Unet (Çiçek et al., 2016) as shown in Fig 7. Our U-net structure consists of contracting and expanding paths. The contracting path consists of the repetition of the following blocks: 3D conv- Instance Normalization (Ulyanov et al., 2016)- ReLU. Here, the generator has symmetric configuration so that both encoder and decoder have the same number of layers, i.e. $\kappa = 7$. Throughout the network, the convolutional kernel dimension is $3 \times 3 \times 3$. There exists a pooling layer and skipped connection for every other convolution operations. To enhance the image contrast, we add additional sigmoid layer at the end of U-Net. On the other hand, the low-resolution image generator from high resolution input is based on a single 3D convolution layer that models a 3D blurring kernel. The size of the 3D PSF modeling layer is chosen depending on the problem set by considering their approximate PSF sizes. In this paper, the size of the 3D PSF layer is set to $20 \times 20 \times 20$.

As for the discriminators, we follow the original CycleGAN that uses multi-PatchGANs (mPGANs) (Isola et al., 2017), where each discriminator has input patches with different sizes used. As shown in Fig 8, it consist of three independent discriminators. Each discriminator takes patches at different sizes: original, and half, and quarter size patches.

### D.2.1 REAL EXPERIMENT

A total 18 epi-fluorescent (EPF) microscopy images of tubulin with a size of $512 \times 512 \times 30$ were used for training, and one for validation. As for unmatched sharp image volume, we used deblurred image generated by utilizing a commercial software AutoQuant X3 (Media Cybernetics, Rockville). The EPF volume depth was increased to 64 using the reflection boundary condition. Due to GPU memory limitations, the EPF volume was split into $64 \times 64 \times 64$ size patches. For data augmentation, rotation, flip, translation, and scale were imposed on the input patches. We normalized the patches and set them to [0,1]. Adam optimizer (Kingma & Ba, 2014) was also used for training. The learning rate was initially set to 0.0001, which is decreased linearly after 40 epoch, and the total number of epoch was 200 epoch. For the optimizer, we used only a single batch.

|     |                        | PSNR (dB)   | SSIM       |
| --- | ---------------------- | ----------- | ---------- |
|     | Input                  | 18.1087     | 0.5651     |
| (a) | Supervised learning    | 25.6607     | 0.9209     |
|     | Conventional CycleGAN  | 25.7616     | 0.9444     |
|     | Proposed CycleGAN      | **26.4960** | **0.9513** |
|     | Input                  | 17.8018     | 0.5201     |
| (b) | Supervised learning    | 25.4048     | 0.9162     |
|     | Conventional CycleGAN  | 25.5837     | 0.9410     |
|     | Proposed CycleGAN      | **26.2891** | **0.9480** |
|     | Input                  | 17.5045     | 0.4929     |
| (c) | Supervised learning    | 25.1191     | 0.9113     |
|     | Conventional CycleGAN  | 25.3379     | 0.9369     |
|     | Proposed CycleGAN      | **26.0044** | **0.9437** |

Table 2: Performance comparison of various methods of Fig. 9 in terms of PSNR and SSIM.

### D.2.2 SIMULATION STUDY

For simulation studies with the ground-truth data, we used synthetic microtubule network data set (Sage et al., 2017) to train and validate our model. Specifically, from the ground-truth high resolution synthetic microtubule images, we generate blurred images by convolving them with a model PSF. In particular, the numerical PSF was computed using the Born and Wolf model (Kirshner et al., 2011), which is given by

$$h(r_x, r_y, r_z) = \left| C \int_0^1 J_0 \left[ k \frac{NA}{n_i} \rho \sqrt{r_x^2 + r_y^2} \right] e^{-\frac{1}{2} jk\rho^2 r_z \left( \frac{NA}{n_i} \right)^2} \rho d\rho \right|^2 \tag{28}$$

where $C$ is a normalizing constant, $k = 2\pi/\lambda$ is the wave number of emitted light, $\lambda$ is the wavelength, $NA$ is the numerical aperture, and $n_i$ is the refractive index of immersion layer. For all simulation, we use $NA = 1.4$, and $n_i = 1.5$. Then, the simulation is performed with different $\lambda$ for the PSF model (28). The convolution was performed using DeconvolutionLab2 (Sage et al., 2017).

The convolved data was then added with the mixture of zero mean Gaussian with the standard deviation $\sigma = 3$ and Poisson noises with the parameter $m = 7$. The size of the synthetic data was $256 \times 512 \times 128$, and 18 samples of the synthetic data were used for training and the other one is used for validation. Due to memory limitations, the volume was split into $64 \times 64 \times 64$ size patches, which are used as inputs to the neural network.

Fig. 9 shows qualitative comparisons of different reconstruction methods over datasets generated by the theoretical PSFs at different wavelengths. Here, the training was performed using the blurred data with $\lambda = 500$nm PSF kernnel, but the inference was performed using data with wavelength 400 $nm$, 500 $nm$, and 600 $nm$, respectively. The goal of this study is to investigate the generalization power of the proposed method with respect to different wavelengths.

The supervised method removed most blur and noise; however, it lost many fine details of the tubule structure and showed false continuity over the tubule structure. The conventional cycleGAN method showed better qualitative results than the supervised one, presenting finer details and the correct continuity of the tubule structure, but still some of the structures were not fully recovered. On the other hand, the proposed method showed the best qualitative results, recovering most of the microtubule structures. Moreover, as shown in Table 2, the proposed method showed the best f and SSIM scores. This confirmed that the proposed method generalizes well, even when the PSF model for the training data is not matched well with that of the inference phase.

## E  COMPARISON OF THE CLASSICAL AND THE PROPOSED PENALTY FOR INVERSE PROBLEMS

Recall that the inverse problem $y = \mathcal{H}x$ have many solutions due to the ill-poseness of the mapping $\mathcal{H}$. The main motivation of the classical PLS presented in Section 2.2 is to introduce the penalty for choosing the solution $x$, which is most likely based on the prior distribution of data (see Fig. 10(a)). On the other hand, unsupervised learning with training data provides an another way to resolve the ambiguity of the feasible solutions. Specifically, if we define a *single-valued* function $G_\Theta(y)$ and

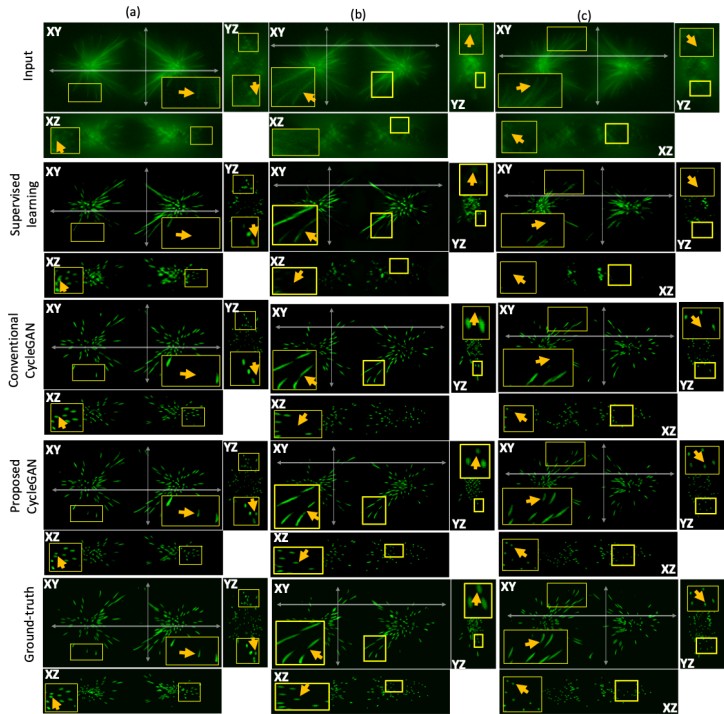

Figure 9: Generalization performance comparison of various methods. The following parameters were used for the training and inference: $NA = 1.4, n_i = 1.5$, and $\sigma = 3, m = 7$. For the training data, the wavelength was $\lambda = 500nm$. (a) Reconstruction result at the inference phase using the data with $\lambda = 400nm$. (b) Reconstruction result at the inference phase using the data with $\lambda = 500nm$. (c) Reconstruction result at the inference phase using the data with $\lambda = 600nm$. Two different views (YZ, XZ) were displayed along the corresponding lines. The ROIs (marked yellow) show the area for the enlarged parts.

impose the constraint $x = G_{\Theta^*}(y)$ with the learned parameter $\Theta^*$, many of the feasible solutions for $y = \mathcal{H}x$ can be cut out as shown in Fig. 10(b). Now, the remaining question is how to determine the optimal parameter $\Theta^*$. This is where the optimal transport theory comes into play. Specifically, our OT formulation tries to find the parameter $\Theta^*$ by minimizing the average transportation cost $c(x, y; \mathcal{H}, \Theta^*)$ so that it eventually approaches zero, the primal and the dual solution may become equivalent, and $G_{\Theta^*}$ can be an inverse of the forward operator $\mathcal{H}$. This is the main motivation for using the new penalty function in our formulation.

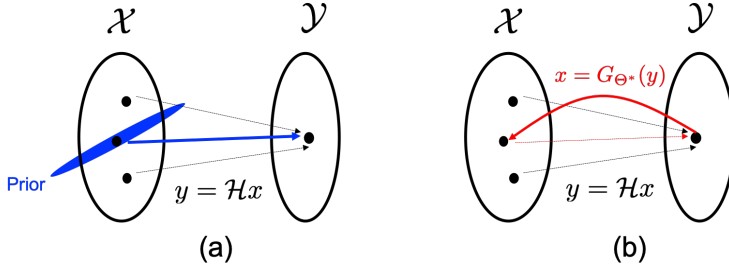

Figure 10: Two strategies for resolving ambiguities in the feasible solutions in an ill-posed inverse problem. (a) Classical PLS approach using a close form prior distribution, and (b) our PLS approach using an inverse mapping to define a prior.

