# OpenReview forum: "OPTIMAL TRANSPORT, CYCLEGAN, AND PENALIZED LS FOR UNSUPERVISED LEARNING IN INVERSE PROBLEMS"
_ICLR.cc/2020/Conference — Reject_

### Official Review · AnonReviewer1 · 2019-10-18
**Official Blind Review #1**

**Rating:** 6

**Review:**

The paper presents an interesting connection between cycleGAN, penalized Least Squares (PLS) and optimal transport (OT). From the PLS with deep regularization formulation, the authors cast a general OT problem with a specific cost combining measurement and reconstruction errors (eq. 9). From this formulation, the problem is expressed as a combination of a cycle consistency loss and an OT loss. The formulation is more generic than the classical cycleGAN formulation. Qualitative experiments are conducted on two different scenarii: accelerated MRI and deconvolution microscopy, for which the proposed method achieves good performances.

In general I like the paper and the corresponding idea. However, I had a hard time understanding some of the key elements of the proposed method. Notably, in lemma 1, it is hard to understand in which case
\mathcal{X} \times \mathcal{X} / A \union B is not an empty set. It is clearly the case whenever G or H are bijections, or whenever an exact reconstruction is achievable, but I guess a more open discussions could be conducted. Also, the definition of A and B are clearly dependent of G and H and this could be highlighted in the notation. As noted in proposition 2, however, the corresponding OT loss can be computed on the entire subset X and Y (since the Kantorovich potentials match those obtained on the restricted set). As a consequence, we can see the overall training procedure when H is fixed (as it is the case in both experiments), as learning for a generator in a WGAN way, that also enforces a consistency constraint (that can be seen as a regularization of the OT problem). While I like this idea, I think there might be some better justification for it (it all starts with Eq. 9 and the choice of this model. Why enforcing this minimal cost equation ?). Note that the model imposes that p=q=1, which is somehow limited. I guess other choices of p and q could be possible, eventually using regularized version of OT to remove the hard constraints on the kantorovich potentials

I am willing to revise my note positively provided that a sufficient number of convincing explanations are given on my previous remarks.

Minor remark
Eq. 8 should include a weighting factor between the two terms since the dimensions are not the same.
In the proof of Lemma 1, since you derive an equality for T, do you have to establish the first inequality ?
Prop. 1 I guess there is an error on the first line (the integral is not over \mathcal{X} \times \mathcal{X} / A \union B given the previous definition
Missing related work
DLOW: Domain Flow for Adaptation and Generalization Rui Gong, Wen Li, Yuhua Chen, Luc Van Gool; The IEEE Conference on Computer Vision and Pattern Recognition (CVPR), 2019, pp. 2477-2486



**Experience Assessment:**

I have published one or two papers in this area.

**Review Assessment: Checking Correctness Of Derivations And Theory:**

I carefully checked the derivations and theory.

**Review Assessment: Checking Correctness Of Experiments:**

I assessed the sensibility of the experiments.

**Review Assessment: Thoroughness In Paper Reading:**

I read the paper thoroughly.

---

> ### Author Response · Authors · 2019-11-10
> **Thanks for your constructive comments**
>
> [Q] The paper presents an interesting connection between cycleGAN, penalized Least Squares (PLS) and optimal transport (OT)...  The formulation is more generic than the classical cycleGAN formulation.
>
> ==>  Thanks for your careful reading and understanding of our contribution.
>
> [Q] In general I like the paper and the corresponding idea. However, I had a hard time understanding some of the key elements of the proposed method. Notably, in lemma 1, it is hard to understand in which case
> \mathcal{X} \times \mathcal{X} / A \union B is not an empty set. It is clearly the case whenever G or H are bijections, or whenever an exact reconstruction is achievable, but I guess a more open discussions could be conducted.
>
> ==> Thank you for your constructive comments. We would like to remind you of the nature of quantities A and B. Since both are graphs of functions, they are low-dimensional manifolds embedded in the $X \times Y$ ambient space. Accordingly, unlike the reviewer's concern, the technical difficulty arises as the relative measure of $A\cup B$ may become zero. However, depending on the existence of the singularity in the distribution, this term can contribute a significant portion of the total T loss. In this paper, we claim that this situation leads to the architecture of cycleGAN.
> To clarify this, we have elaborated on our discussion on the singularity in the distribution on Page 4 and 5.  Moreover, the dependency on the singularity is also explicitly shown in $\ell_{OT}$  or $\ell_{GAN}$ so that how these terms behave depending on $\sigma$ and $\rho$.
>
> [Q]  Also, the definition of A and B are clearly dependent of G and H and this could be highlighted in the notation. As noted in proposition 2, however, the corresponding OT loss can be computed on the entire subset X and Y (since the Kantorovich potentials match those obtained on the restricted set).
>
> ==> Thank you for the excellent suggestion.  We have revised the notation accordingly.
>
> [Q]  As a consequence, we can see the overall training procedure when H is fixed (as it is the case in both experiments), as learning for a generator in a WGAN way, that also enforces a consistency constraint (that can be seen as a regularization of the OT problem). While I like this idea, I think there might be some better justification for it (it all starts with Eq. 9 and the choice of this model. Why enforcing this minimal cost equation ?).
>
> ==> Thank you for your constructive comments.  We appreciate your comments that the consistency constraint works as a regularization term of the OT problem.  Accordingly,  we have employed your point in this revision.  Moreover, instead of claiming that our method is a simple extension of the existing PLS with deep learning prior, we significantly expanded the discussion on the existing PLS approaches to highlight the difference of our formulation.   We emphasize that the reason we enforce these minimal costs is to lead the primal solution to become a true inverse of the forward operator. In addition, we showed that the dual solution in the global optimum corresponds to the primal solution, which is not the case with the standard CycleGAN.   Additionally, we show that while the existing PLS with the deep image prior can be similarly converted to the Kantorovich double formulation (see Appendix C), but it does not provide a simple feedforward neural network. On the other hand, our new formulation leads to a CycleGAN architecture, which leads to a simple neural feed-forward network for the estimation of x. We highlighted this as one of the motivations for our framework.
>
> [Q] Note that the model imposes that p=q=1, which is somehow limited. I guess other choices of p and q could be possible, eventually using regularized version of OT to remove the hard constraints on the kantorovich potentials .
>
> ==> Thanks for your constructive comments. We have added the related discussion on page 6 of the revised paper.
>
> [Q] Eq. 8 should include a weighting factor between the two terms since the dimensions are not the same.
>
> ==> Thank you for your constructive comments. We have revised the notation in  (9). However, in all experimental scenarios in this article, the dimensions of x and y are the same. Therefore, for the sake of simplicity, the remainder of the paper is assumed  $\lambda = 1$.
>
> [Q] In the proof of Lemma 1, since you derive an equality for T, do you have to establish the first inequality ?
>
> ==> Thanks for your comments, and we now revised the proof according to your suggestion.
>
> [Q] Prop. 1 I guess there is an error on the first line (the integral is not over \mathcal{X} \times \mathcal{X} / A \union B given the previous definition
>
> ==> We truly appreciate your careful reading and correct all typo you pointed out.
>
> [Q] Missing related work
>
> ==>  Reference added.

---

### Official Review · AnonReviewer2 · 2019-10-23
**Official Blind Review #2**

**Rating:** 6

**Review:**

This paper frames and contextualizes CycleGAN as a stochastic generalization of penalized least squares for inverse problems, providing several unifying theorems, rederiving some modern CycleGAN architectures within the optimal transport framework, and also demonstrating the practical use of modified architectures derived using this framework for accelerated MRI and microscopy.

While I did not check the proofs in detail, the additional generalization of a well-known, working architecture and additional variants used practically seem significant to me. Demonstrating the deep relationships in the proofs and propositions shown here, followed by two clear and concise derivations *and their practical application* is compelling, making this paper broadly applicable to both practitioners of compressed sensing and generative modeling. Frankly, it will take me some time to digest the proofs and overall relationships shown in the Propositions here, as I am not deeply familiar with optimal transport. The applied sections are direct, with Figure 5 being especially meaningful.

There are a few small grammatical issues in the text - a careful re-read and edit with particular focus on grammar and style would be beneficial, though as it stands these small flaws didn't meaningfully detract from the paper.

The primary thing the authors could do in order to raise my score, would be to take an additional pass at grammatical clarity for the paper. More experiments are always beneficial, and I would encourage the authors to release source code to replicate some form of their experiments if possible.

Some additional references which may be useful additions, primarily for interested readers to gain further background on the use of deep models for compressed sensing:
Compressed Sensing with Deep Image Prior and Learned Regularization https://arxiv.org/abs/1806.06438
Compressed Sensing using Generative Models https://arxiv.org/abs/1703.03208
Deep Compressed Sensing http://proceedings.mlr.press/v97/wu19d/wu19d.pdf

**Experience Assessment:**

I do not know much about this area.

**Review Assessment: Checking Correctness Of Derivations And Theory:**

I did not assess the derivations or theory.

**Review Assessment: Checking Correctness Of Experiments:**

I assessed the sensibility of the experiments.

**Review Assessment: Thoroughness In Paper Reading:**

I read the paper at least twice and used my best judgement in assessing the paper.

---

> ### Author Response · Authors · 2019-11-10
> **Thank you for your understanding of our contribution**
>
> [Q] the additional generalization of a well-known, working architecture and additional variants used practically seem significant to me. Demonstrating the deep relationships in the proofs and propositions shown here, followed by two clear and concise derivations *and their practical application* is compelling, making this paper broadly applicable to both practitioners of compressed sensing and generative modeling.  The applied sections are direct, with Figure 5 being especially meaningful.
>
> ==> Thanks for your careful reading and understanding of our contribution.
>
> [Q] There are a few small grammatical issues in the text - a careful re-read and edit with particular focus on grammar and style would be beneficial, though as it stands these small flaws didn't meaningfully detract from the paper.
>
> ==> Thanks for the suggestion. An English native speaker has proofread the revision.
>
> [Q] The primary thing the authors could do in order to raise my score, would be to take an additional pass at grammatical clarity for the paper. More experiments are always beneficial, and I would encourage the authors to release source code to replicate some form of their experiments if possible.
>
> ==> An English native speaker has proofread the revision.  In the revised paper, we have conducted additional experiments and provided quantitative results for both accelerated MRI and deconvolution microscopy. In order to follow the policy stating that  " If you want to share your code with reviewers and ACs confidentially, please use the comment feature after the submission deadline", the anonymized code link is now provided to reviewers and ACs confidentially using the comment feature. Once the paper is accepted, we will make the link public.
>
> [Q] Some additional references which may be useful additions, primarily for interested readers to gain further background on the use of deep models for compressed sensing:
>
> ==>  Reference added.

---

### Official Review · AnonReviewer4 · 2019-11-01
**Official Blind Review #4**

**Rating:** 1

**Review:**

In this paper, the authors present two contributions:
1)	The primary contribution is to show that CycleGAN can be formulated as a probabilistic version of a particular penalized-least squares problem (theory)
2)	As proof of concept, they apply their version of CycleGAN to accelerated MRI and deconvolution microscopy (application)

While I find the idea to be potentially interesting, the presentation of the theory is unclear and not well-motivated; after reading, I’m not convinced that the connection to CycleGAN is as significant as the authors claim. The experimental results are preliminary. My decision is to reject. Below are separate critiques on the sections.

Section 2-3: Hope the authors could clarify / strengthen these points in revision:
-	Since the discussion in Section 3 is based on the optimization problem in Equation (7), this problem should be well-motivated. Currently it is presented as a problem that has been explored previously by Zhang et al and Aggarwal et al. However, after taking a look at those papers, I don’t understand where this regularization term comes from. In these papers, the regularization term (i.e. equation 2 or 3 of Zhang et al) appears independent of y. Since this term is key to the paper, it should be well explained here. E.g. at the end of section 2.2: G_\theta(y) is a CNN pretrained on what task?

-	In the inverse problem, the objective is to estimate x from y. Therefore we care about \argmin x in Equation (7). In the probabilistic setting presented in Equation (8), analogously the objective is to estimate \pi^*, which is the solution to the primal problem. The theory shows that the primal formulation in Equation (8) is equivalent to the dual formulation in Equation (16), but does not show how the dual solution yields the primal solution, which is lacking as obtaining the primal solution seems to be the point of solving the PLS problem. (Interestingly, in Section 4, the authors are using the dual solution x = G_\theta(y) as if it is the mapping given by pi(x|y)… this needs to be explained.)

-	The authors claim that Proposition 1 shows that the cyclic loss term in their dual formulation is a more general version of the cycle-consistency loss in CycleGAN. But looking closely at Proposition 1 and its proof, it seems that the equivalence holds only for specific weights, not for arbitrary weights. Additionally, the specific weights are unknown (they depend on the solution \pi^* to the primal problem…). I do not understand the claim that this is a generalization of cycle-consistency loss, nor do I see how the authors implement their version of the cyclic loss as it depends on unknown weights.

-	The connection to CycleGAN seems to hold only when p=q=1?

-	End of section 3: The authors conclude “our cost formulation using (17) with (18) and (19) is more general compared to the standard CycleGAN, since a general form of measurement data generator Hx can be used”. I don’t see the connection between the theory and this claim. Even with CycleGAN, both generators can be arbitrary or fixed if one of them is known.

-	The proofs are easy to follow, though perhaps they could be moved to the Appendix in favor of providing more motivation and explanation in the main text.

Section 4:
-	The authors motivate the problem with the PLS setup but then they use the learned regularization term x = G_\theta(y) as if it is the mapping given by pi(x|y).  I am confused by this.
-	Putting aside the connection to the PLS problem, my interpretation of the experimental setup is that the authors use CycleGAN with Wasserstein GAN loss instead of the classic discriminator loss, where one of the generators is known (and hence only one generator/discriminator pair is needed). I might be missing something, but I’m not sure that this approach is different enough from CycleGAN.
-	Considering that the authors have the ground truth, they could provide quantitative evaluation of their method against other methods, rather than showing a few qualitative results where it is working.


**Experience Assessment:**

I have published one or two papers in this area.

**Review Assessment: Checking Correctness Of Derivations And Theory:**

I assessed the sensibility of the derivations and theory.

**Review Assessment: Checking Correctness Of Experiments:**

I carefully checked the experiments.

**Review Assessment: Thoroughness In Paper Reading:**

I read the paper thoroughly.

---

> ### Author Response · Authors · 2019-11-10
> **Our contribution has been highlighted**
>
> [Q] .. I’m not convinced that the connection to CycleGAN is as significant as the authors claim...
>
> ==>  We would like to assure the reviewer that the primary motivation of this work is to provide a principled method for designing cycleGAN for various inverse problems by using the original physics as regularization for OT. This is an important advance over the existing CycleGAN, which is mainly derived from trial and error. In particular,  the reason we enforce the proposed PLS cost for the OT problem is to lead the primal solution to become a true inverse of the forward operator at the global minimum. As the other reviewers pointed out, our formulation is more generic than the classical cycleGAN formulation, and demonstrating their deep relationships, followed by two clear and concise derivations and their practical application,  is compelling.  See General Comments.
>
> [Q]  ...  Currently it is presented as a problem that has been explored previously by Zhang et al and Aggarwal et al. H.. Since this term is key to the paper, . ..
>
> ==> We found that the latex bib file incorrectly referred to different Zhang's paper. We are quoting a correct one now. However, we understand the reviewers' concern that the existing deep learning prior approaches are indirectly related to y.  In this revision, rather than emphasizing our method as an extension of the existing PLS with deep learning prior, we have highlighted the difference in our formulation. We emphasize that the reason for using (9) is to lead the primal solution to become a true inverse of the forward operator.
>
> [Q]...   but does not show how the dual solution yields the primal solution....
>
> ==> Thanks to our PLS formulation (9),  we confirmed that the dual solution and the primal solution are equivalent when the global optimum is achieved with $c(x, y; H, \Theta) = 0$.
>
> [Q] The authors claim ... the cyclic loss term in their dual formulation is a more general version of the cycle-consistency loss in CycleGAN. ...
>
> ==> Kindly note that the term "general version" is used when, under certain conditions, the new formulation can be reduced to the standard one. Similar to the standard cycleGAN, the hyperparameters should be selected by trial and error. We agree that this is the limitation for both the standard and the proposed cycleGAN. However, our "generalized" formulation gives better insight into the selection of hyperparameters. In fact, the parameter is the relative match between two data distributions. If it turns out that both pairs are perfect, the parameter should be 1. In a real training scenario, the perfect match can not be found so that the hyperparamereter should be between 0 and 1.
>
> [Q] The connection to CycleGAN seems to hold only when p=q=1?
>
> ==> We only considered  $p = q = 1$ due to the simple c-transformation. The widely used W-GAN is derived similarly by assuming p = 1. The use of general PLS costs is of course possible and could lead to an interesting variation of the cycleGAN architecture.
>
> [Q] ... Even with CycleGAN, both generators can be arbitrary or fixed if one of them is known.
>
> ==> The standard CycleGAN could use a fixed generator, but there is no optimal design criterion to show why this is better. On the other hand, our formulation requires only a single deep generator if the measurement physics is given by the forward model Hx. In fact, as one of the other reviewers noted,  this can be seen as a consistency term from the forward model to acts as a regularization term for OT.
>
> [Q]The proofs ... could be moved to the Appendix..
>
> ==> Done.
>
> [Q] ... but then they use the learned regularization term x = G_\theta(y) as if it is the mapping given by pi(x|y).
>
> ==> Our novel PLS cost (9) enforces the dual solution to be the primal solution of the PLS when the global minimum is reached. Therefore, in this case, $x = G_\Theta (y)$ is actually the map given by $\pi (x|y)$.
>
> [Q] ...my interpretation of the experimental setup is that the authors use CycleGAN with Wasserstein GAN loss instead of the classic discriminator loss, where one of the generators is known ...  but I’m not sure that this approach is different enough from CycleGAN.
>
> ==> Our main contribution is the principal derivation of the cycleGAN architecture for various inverse problems. If we use p = q = 1, the resulting discriminator loss becomes Wasserstein GAN. However, with different p, q values and the regularized version of optimal transport, our formulation offers a new class of discriminator architecture. In addition, if the forward mapping is unknown, the framework is reduced to the standard cycleGAN with two deep generators.  Therefore, our framework is much more flexible.
>
> [Q] ..they could provide quantitative evaluation of their method against other methods...
>
> ==> Done. In the revised paper, we have provided the quantitative results for both accelerated MRI and deconvolution microscopy. The results clearly showed the advantages of the proposed method.

---

> > ### Comment · AnonReviewer1 · 2019-11-13
> > **G_\theta(y) is not the exact mapping of pi(x|y)**
> >
> > I am reviewer #1. I do actually agree with the previous comment from reviewer 4:
> >
> > [Q] ... but then they use the learned regularization term x = G_\theta(y) as if it is the mapping given by pi(x|y).
> >
> > [A] ==> Our novel PLS cost (9) enforces the dual solution to be the primal solution of the PLS when the global minimum is reached. Therefore, in this case,  $G_\theta(y)$ is actually the map given by $pi(x|y)$.
> >
> > I believe it would be true only if the optimization of G were carried over the whole set of 1-Lipschitz functions,
> > which is not the case since you are optimizing over the set of parameters for a given class of neural networks. Here, the WGAN formulation is actually a lower bound of the true optimal transport. Can the authors comment on that ?

---

> > > ### Author Response · Authors · 2019-11-13
> > > **We have already considered this situation**
> > >
> > > Thank you for your constructive comments. We would like to remind the reviewer that $G_\Theta(y)$ is not a discriminator, so it is not necessary to be a 1-Lipschitz function.
> > >
> > > However, due to the limited capacity of the neural network $G_ \Theta$, we agree that the global minimum may not be reached and the dual solution may not be equal to the primal one. In our revised article, we have already considered this limitation. See page 3-4 “In practice, due to the limited capacity of the neural network, the global minimum with $c(x, y;_\Theta, H) = 0$ may never be achieved. Even in this case, thanks to the symmetric form of PLS, the cost in (9) has an important implication: the neural network $G_\Theta$ is now estimated by enforcing the consistency $y = Hx$ as a regularization term..”
> > >
> > > However, we believe that this does not limit the benefits of our dual formulation. It should be remembered that even in the existing W-GAN, the dual formulation is equivalent to the primal optimal transport problem only if the maximization of the discriminator is taken for all sets of 1-Lipschitz functions. This is not the case in practice, because the standard W-GAN is optimized over the parameter set for a particular class of neural networks. Nevertheless, the existing W-GAN provides very realistic images and leads to many interesting extensions that were not possible with the primal formulation of optimal transport.  We have observed the same benefits of our cycleGAN formulation.

---

> > > > ### Comment · AnonReviewer1 · 2019-11-13
> > > > **Thanks for the clarification**
> > > >
> > > > Indeed I was too quick in saying that G instead of the discriminator needed to be 1-Lipschitz, but you got my point right. Thanks for this clarification.

---

> > ### Comment · AnonReviewer4 · 2019-11-14
> > **Paper is improved after revision and comments; additional questions and comments below**
> >
> > Overall, I greatly appreciate the improvements that the authors have made to their paper. After reading the revised work and the comments in the discussion, my assessment of the contributions have been clarified to the following:
> >
> > 1)	The authors propose a *new* PLS problem with a penalty function based on the learned mapping G(y). (Before my impression was that the author’s contribution was to connect an existing PLS formulation to CycleGAN that I could not find evidence of in the cited work).
> > 2)	The authors show that their proposed PLS problem can be reformulated in a way that the CycleGAN framework can be applied for optimization.
> > 3)	The authors apply their method for optimizing the proposed PLS problem based on the CycleGAN framework to practical problems. They now also provide quantitative metrics and baselines for comparing their PLS solution to other solutions. (Before my impression was that the approach was claimed to be superior and/or sufficiently different from CycleGAN. Now, my understanding is that they solve the PLS problem using the CycleGAN framework. Both the revised paper and the comments in the discussion have clarified this for me.)
> >
> > Given my improved understanding of the work, I now have the following comments:
> >
> > -	What is the motivation for proposing the new PLS formulation with the penalty ||G(y) – x|| where G is learned? In the prior work presented in section 2.2, for example, the motivation/intuition is along the lines of (1) we want to solve the inverse problem y = Hx, but due to the non-invertible nature of the mapping H, there may be many multiple solutions, so therefore (2) we introduce the penalty to choose the solution x that is most likely based on the prior distribution of data in the training set or a physical model given by Q. Providing some motivation for the proposed penalty function in connection to the inverse problems the authors are interested in solving can strengthen the significance of the work and motivate the exposition in the beginning of Section 3, as otherwise it seems like the penalty in the PLS problem is chosen arbitrarily for mathematical convenience in the derivation.
> >
> > -	With regards to using x = G(y) as if it is the mapping given by pi(x|y), I still do think there is a gap between the theory and the application to inverse problems that goes beyond whether or not G has sufficient capacity when parameterized using neural networks. Based on the introduction of inverse problems and the application in the experimental section, it seems that often H produces some sort of “low-resolution” measurement of y from x and is therefore not invertible, which makes the inverse problem ill-posed as the authors state. Is it the case then that pi(x|y) is not deterministic and that c=0 cannot be achieved, even if G is given infinite capacity? It would be great if the authors could comment on this.
> >
> > -	To add on to both comments above, based on a couple of factors (e.g., that the authors are using x = G(y) as the inverse mapping in the applications, the sentence (“the neural network is now estimated by enforcing the consistency y=Hx as a regularization term”) in Section 3, and the statement that when c=0, G is exactly the inverse of H) , it seems that a more convincing exposition the authors could be going for is informally along the lines of: (1) we want to estimate the inverse map from y to x, so let G represent the estimated mapping and directly optimize ||G(y) – x|| with the regularization term ||y – Hx|| to enforce the consistency of the solution;  (2) this makes the problem take on the form of a PLS problem with ||G(y) – x|| as the penalty term, (3) BUT since we don’t actually know the error ||G(y)-x|| due to not having coupled data, we take the solution G that minimizes the error over with respect to the optimal coupling of the distributions of x and y, leading to the optimal transport problem shown in Equation (11), which can be optimized using the CycleGAN framework. Could the authors comment on whether this interpretation is appropriate?
> >
> > -	The quantitative results in the experimental section are appreciated. It would be helpful to include in the caption a quick description of what the metrics used are. Additionally, it could greatly strengthen the experimental section (time permitting) to compare to a different method that solves these types of inverse problems, since now I understand that the authors' contribution is a *new* formulation of the PLS problem that is optimized using CycleGAN framework.

---

> > > ### Author Response · Authors · 2019-11-14
> > > **We appreciate your  comments,  and have addressed them in a new revision.**
> > >
> > > [Q] Overall, I greatly appreciate the improvements that the authors have made to their paper. After reading the revised work and the comments in the discussion, my assessment of the contributions have been clarified to the following:
> > >
> > > ==> Thanks for your understanding of our contributions.
> > >
> > > [Q] What is the motivation for proposing the new PLS formulation with the penalty ||G(y) – x|| where G is learned? .. as otherwise it seems like the penalty in the PLS problem is chosen arbitrarily for mathematical convenience in the derivation.
> > >
> > > ==>  Thanks for your constructive comments.  See Appendix E in the new revision.    Recall that the inverse problem $y =\mathcal{H} x$ has many solutions due to the ill-posedness of the mapping $\mathcal{H}$. The main motivation of the classical PLS presented in Section 2.2. is to introduce the penalty for choosing the solution $x$, which is most likely based on the prior distribution of data (see Fig. 10(a) in Appendix E). On the other hand,  unsupervised learning with training data provides another way to resolve the ambiguity of feasible solutions. Specifically, if we define a single-valued function $G_\Theta(y)$ and impose the constraint $x=G_{\Theta^*}(y)$ with the learned parameter $\Theta^*$, many of the feasible solutions for $y=\mathcal{H} x$ can be cut out  as shown in Fig.10(b) in Appendix E.  Now, the remaining question is how to determine the optimal parameter $\Theta^*$. This is where the optimal transport theory comes into play. Specifically, our OT formulation tries to find the parameter $\Theta^*$ by minimizing the average transportation cost. Then, the resulting cost $c(x,y;\mathcal{H},\Theta^*)$ can eventually approach zero, the primal and the dual solution may become equivalent, and $G_{\Theta^*}$ can be an inverse of the forward operator $\mathcal{H}$. This is the main motivation for using the new penalty function in our formulation.
> > >
> > > [Q] With regards to using x = G(y) as if it is the mapping given by pi(x|y), I still do think there is a gap between the theory and the application to inverse problems ... Is it the case then that pi(x|y) is not deterministic and that c=0 cannot be achieved, even if G is given infinite capacity? It would be great if the authors could comment on this.
> > >
> > > ==> Thanks for your very insightful comments. As shown in Fig. 10 in Appendix E, due to the ill-posedness of an inverse problem, for the same “low-resolution” measurement, there may exist multiple feasible solutions, among which one is the true solution.  Thus,  a single-valued neural network $G_{\Theta^*}(y)$ with an infinite capacity can always find the unique optimal solution. A practical limitation, however, is that an infinite number of training data is required to ensure that the optimal neural network parameter $\Theta^*$ is found.  With the limited number of training data, $c = 0$ may not be achieved, and in this case we agree that $x = G_{\Theta} (y)$ is an approximation of $\pi (x|y)$.
> > >
> > > [Q] ..  it seems that a more convincing exposition the authors could be going for is informally along the lines of: ... Could the authors comment on whether this interpretation is appropriate?
> > >
> > > ==> Thanks for your summary of our contribution. Yes, your understanding is correct.  If you think it necessary, we will gladly use your summary exposition in conclusion or introduction.
> > >
> > > [Q] The quantitative results in the experimental section are appreciated. It would be helpful to include in the caption a quick description of what the metrics used are. Additionally, it could greatly strengthen the experimental section (time permitting) to compare to a different method that solves these types of inverse problems, since now I understand that the authors' contribution is a *new* formulation of the PLS problem that is optimized using CycleGAN framework.
> > >
> > > ==> Done. We have added the description of the metrics in the caption of Fig. 3.  In Fig.  5, we have also added the deconvolution results using the commercially available software AutoQuant (Media Cybernetics, Rockville), which is based on the iterative deconvolution method for a classical PLS problem. The comparative results confirmed that only the proposed method maintained the structural connectivity of the microtubule in the cell.

---

### Author Response · Authors · 2019-11-10
**General Comments to the Reviewers**

We thank all reviewers for their constructive comments. In general, reviewers acknowledged that this paper presents an interesting connection between cycleGAN, PLS, and OT. The reviewers also agreed that our formulation is more generic than the classical cycleGAN formulation. They also mentioned that demonstrating their deep relationships, followed by two clear and concise derivations and their practical application,  is compelling,  making this paper broadly applicable to practitioners.

In this revision, we emphasized that the primary motivation for this work is to provide a principled method for designing cycleGAN for various inverse problems by using the original physics as regularization for OT. This is an important advance over the existing CycleGAN, which is mainly derived from trial and error. We emphasize that the reason we enforce the proposed PLS cost (9) for the OT problem is to lead the primal solution to become a true inverse of the forward operator. In addition, we showed that the dual solution in the global optimum corresponds to the primal solution, which is not the case with the standard CycleGAN.

To further clarify our contribution and address the reviewers' comments,  the following major changes have been made.

[1]  In Section 2.2, we have significantly broadened the discussion on existing PLS approaches with deep learning prior,  before highlighting the difference in our formulation. We also showed that the existing Deep Image Prior (DIP) model can be also converted to a similar Kantorovich dual formulation (see Appendix C), but it does not provide a simple feedforward neural network to the estimation of $x$. On the other hand, our new formulation leads to a CycleGAN architecture, which leads to a simple neural feed-forward network for the estimation of $x$.

[2] We have significantly enriched the experiments and provided quantitative comparison results for both accelerated MRI and deconvolution microscopy. The results clearly showed the advantages of the proposed method over the existing cycleGAN approaches.

[3] To avoid potential confusion of the readers, we have provided extensive discussion on the nature of set A and B in Section 3. We emphasized that both are graphs of functions, which are low-dimensional manifolds embedded in the $X\times Y$ ambient space. Therefore, their relative measure can be zero, but depending on the existence of the singularity in the distribution, this term can contribute a significant portion of the total T loss and leads to the architecture of cycleGAN. This geometric feature and implications have been described in detail. Moreover, the dependency on the singularities of the GAN loss term is now shown explicitly.

[4] We concluded the theoretical sections by emphasizing the following contributions:

* The standard cycleGAN usually requires two generators in the form of deep neural networks. On the other hand, our formulation often requires a single generator using a deep neural network, if the other generator is given by the forward model Hx. This significantly reduces the number of weights, which makes the training much more stable. This can be viewed as the consistency term from the forward model working as a regularization term for OT.

*  When the solution of the dual problem achieves the global minimum with diminishing cost, then the dual solution $x = G_\Theta(y)$ is equivalent to the primal solution (10), validating the dual approach. This is not the case in the conventional cycleGAN with two deep generators.

*  While we only consider $p = q = 1$ due to the simple c-transform, the use of the general PLS cost would be interesting, and it may lead to an interesting variation of the cycleGAN architecture. This could be done using a regularized version of OT.

* The weight parameters in (1) and (20) originate from the singularity of the optimal joint measure. However, the estimation of weight parameters is not feasible in practice, so the constant hyperparameters in (21) are usually used, and the hyperparameters should be selected by trial and error, which is the same as in the standard cycleGAN. Nevertheless, the probability interpretation may lead to a more simplified search for the hyper-parameters.

[5] This paper has been proofread by a professional English editor.

---

### Decision · Program_Chairs · 2019-12-19

**Decision:**

Reject

**Comment:**

This paper provides a novel approach for addressing ill-posed inverse problems based on a formulation as a regularized estimation problem and showing that this can be optimized using the CycleGAN framework. While the paper contains interesting ideas and has been substantially improved from its original form, the paper still does not meet the quality bar of ICLR due to a critical gap between the presented theory and applications. The paper will benefit from a revision and resubmission to another venue.